# Is Cardiorespiratory Fitness Independently Associated with Fatigue in Patients with Transient Ischemic Attack or Minor Stroke?

**DOI:** 10.3390/brainsci13040561

**Published:** 2023-03-27

**Authors:** Inger A. Deijle, Erwin E. H. Van Wegen, Renske M. Van den Berg-Vos, Gert Kwakkel

**Affiliations:** 1Department of Neurology, OLVG Hospital, 1061 AE Amsterdam, The Netherlands; i.deijle@olvg.nl (I.A.D.);; 2Department of Quality and Improvement, OLVG Hospital, 1061 AE Amsterdam, The Netherlands; 3Department of Rehabilitation Medicine, Amsterdam UMC, Location Vrije Universiteit Amsterdam, Boelelaan 1117, 1081 HZ Amsterdam, The Netherlands; 4Rehabilitation and Development, Amsterdam Movement Sciences, Amsterdam UMC, Location Vrije Universiteit Amsterdam, 1081 HZ Amsterdam, The Netherlands; 5Amsterdam Neurosciences-Neurovascular Disorders, Amsterdam UMC, Location AMC, 1105 AZ Amsterdam, The Netherlands; 6Department of Neurology, Amsterdam UMC, Location AMC, 1105 AZ Amsterdam, The Netherlands; 7Department of Physical Therapy and Human Movement Sciences, Northwestern University, Chicago, IL 60611, USA; 8Department of Neurorehabilitation, Amsterdam Rehabilitation Research Centre, Reade, 1054 HW Amsterdam, The Netherlands

**Keywords:** stroke, fatigue, fitness

## Abstract

Fatigue is a common complaint and a disabling symptom among patients following transient ischemic attack (TIA) or minor stroke. In patients with stroke, decreased cardiorespiratory fitness (CRF) is believed to be related to increased severity of post-stroke fatigue (PSF). However, this association between PSF and CRF in patients with TIA or minor stroke has been less investigated, and currently there is no proven treatment for PSF. We aimed to determine the association between PSF and CRF in patients with TIA or minor stroke and to find out whether this association was distorted by confounders. A cross-sectional association study was conducted among a total of 119 patients with TIA or minor stroke. PSF was measured by the Fatigue Severity Scale (FSS) and CRF was quantified by maximal exercise capacity (V̇O2max). The FSS showed a significant association with V̇O2max (ß = −0.061, SE: 0.022; *p* = 0.007). This association was confounded by anxiety (ß = −0.044, SE: 0.020; *p* = 0.028) and depression (ß = −0.030, SE: 0.022; *p* = 0.177) as measured by the subscales of the Hospital Anxiety and Depression Scale (HADS). After controlling for HADS scores on depression and anxiety, the univariate relationship between V̇O2max and FSS was no longer significant. These results suggest that the association between PSF and CRF in patients with TIA or minor stroke is weak and significantly confounded by the factors of depression and anxiety.

## 1. Introduction

Fatigue is a common complaint among patients following stroke [1]. The reported prevalence of poststroke fatigue (PSF) ranges from 31% [2] to 68% [1,3], whereas up to 40% of stroke survivors report fatigue as their worst or one of their worst symptoms [4]. Although fatigue has been found to be positively associated with the severity of stroke [5], the majority of patients with transient ischemic attack (TIA) or minor stroke also report fatigue as one of their main disabling symptoms [3]. PSF negatively affects daily functioning and societal participation and has been associated with poor neurologic recovery and higher case fatality [5,6].

PSF is a multifaceted phenomenon with a currently unknown pathophysiology, but it is likely to result from a complex interaction between factors such as cardiorespiratory fitness (CRF) [5,7], demographic factors such as age [8,9] and gender [10,11], physical activity level [5,11], cognitive functioning [5,12], pre-stroke fatigue [13,14], depression [10,11], anxiety [5,10], stroke type [5,10], co-morbidity [10], sleep disturbances [5,10], pain [5,15], and physical deficits [5,11]. Currently, there is no proven effective pharmacological or non-pharmacological treatment for PSF [16,17].

In patients with stroke, decreased CRF is significantly associated with impaired motor function and physical inactivity [5] and it is thought to be associated with increased severity of PSF [18]. CRF is also decreased in patients with TIA or minor stroke. Their levels of physical activity often do not meet physical activity recommendations and are low compared to those of healthy peers [19,20,21]. However, the association between PSF and CRF in patients with TIA and minor stroke has been poorly investigated.

The primary aim of the present cross-sectional association study was, therefore, to investigate the relationship between PSF and CRF in patients with TIA or minor stroke, one year post-onset. We hypothesized that CRF measured at one year would be significantly associated with the self-reported severity of PSF. The secondary aim of the present study was to investigate whether the possible association between PSF and CRF could be distorted by potential confounders that are also related to PSF and CRF. Based on the literature, we hypothesized that age [8,9], gender [10,11], self-reported physical activity level [5,11], cognitive functioning [5,12], depression [10,11], anxiety [5,10], stroke type [5,10], and co-morbidity [10] could be significant factors that might distort the association between PSF and CRF.

## 2. Materials and Methods

### 2.1. Population and Design

The current cross-sectional association study used data from the “MoveIT” trial, a single-center, observer-blinded, randomized controlled trial that examines the effects of a one-year exercise intervention on cognition in patients post-TIA or minor ischemic stroke [19]. The participants in this study were 119 patients who had suffered a TIA or a minor stroke. TIA has been classically defined as a brief (lasting < 24 h) episode of neurological dysfunction caused by focal brain or retinal ischemia and without evidence of acute infarction or tissue injury [22]. Minor ischemic stroke was defined by a National Institutes of Health Stroke Scale (NIHSS) score ≤ 3 [23]. Of the total of 119 patients enrolled in the study, 60 were allocated to the treatment group and 59 were allocated to the control group. The control group received usual stroke care, which consisted of a total of 2–3 follow-up visits to the outpatient clinic for three months after the TIA or minor stroke. During these appointments, patients received motivational interviewing-based counselling, which was focused on the attainment of secondary prevention and the improvement of both lifestyle factors and physical activity. The study procedures were approved by local university and hospital research ethics committees at Amsterdam UMC, Location Vrije Universiteit, NL38008.029.11, and the MoveIT was registered in the Netherlands Trial Register—NL3721. All methods in the MoveIT trial were carried out in accordance with the Declaration of Helsinki. Written informed consent was obtained from all patients. The follow-up period was 2 years after the initial TIA or stroke, and the last patient was assessed in October 2016.

### 2.2. Brief Description of MoveIT Intervention

The MoveIT exercise intervention started with a 12-week exercise program that was performed in groups of 10 patients. The patients received two one-hour sessions of exercise training per week, supervised by two specialized physiotherapists. The exercise program consisted of both aerobic and strength training. The aerobic exercise was performed using a cycle ergometer, treadmill, or rowing machine. The Karvonen formula was used to calculate the target heart rate (THR), as follows: resting heart rate + (% desired intensity × [maximum heart rate derived from the maximum exercise test—resting heart rate]) [24]. Aerobic exercise started at 40% and was gradually increased to 80% THR. During each aerobic exercise training session, heart rates were measured twice to ensure that the patients exercised at the target level. We also used a rating of perceived exertion of 11 to 16 (“light” to “hard”) on the Borg 6–20 Scale [25]. The strength training part of the program, performed on weight machines, started with three sets of 10–12 repetitions at 30% of the repetition maximum, and patients gradually progressed to 60–70%. Patients were instructed and guided to perform home-based exercise sessions three times a week, with the aim of achieving independence in exercising and developing and maintaining active lifestyles that meet recommended physical activity and exercise guidelines. In order to provide insight into the amount and frequency of physical training, patients used an exercise diary, which recorded how much they exercised and what facilitators and barriers they perceived for an active lifestyle. After completion of the 12-week group exercise program, follow-up consisted of three visits to a physiotherapist until 12 months after the diagnosis was made. During these follow-up visits, the patients received counselling from the physiotherapists, based on motivational interviewing. Patients were encouraged to maintain an active lifestyle and to continue exercising, using the exercise diary as an evaluation method.

The MoveIT trial found no additional benefit of a one-year exercise intervention compared to usual care, regarding our primary outcome of global cognitive functioning at one year, as measured with the Montreal Cognitive Assessment (MoCA). In addition, we found no significant between-group differences in secondary outcomes such as cardiorespiratory fitness, the attainment of secondary prevention targets, and self-reported measures of anxiety and depression. The only significant between-group difference was found for fatigue, which was less in the experimental group than in the control group at 12 months. The neutral outcome of our trial suggests that an exercise intervention comprising a 12-week group exercise program and a nine-month follow-up, both under the guidance of specialized physiotherapists, has no favorable effect on global cognitive functioning [19].

### 2.3. Subjects

Patients were eligible if they (1) were at least 18 years old, (2) presented with a TIA or minor ischemic stroke, (3) had experienced the onset of signs and symptoms less than one month ago, (4) were able to walk independently, (5) had been discharged from hospital without need for further rehabilitation, (6) had a Mini-Mental State Examination (MMSE) [26] score ≥ 24, (7) had no aphasia and were able to speak Dutch, (8) had no cardiopulmonary contraindications for physical exercise and exercise testing as outlined by the American College of Sports Medicine [24], and (9) did not suffer from other chronic diseases with an expected survival of less than two years.

### 2.4. Measuring Fatigue

Fatigue was defined as “a subjective feeling of lack of energy, weariness, and aversion to effort” [27]. The Fatigue Severity Scale (FSS), which has been recommended as a valid measurement for patients with stroke [28], was used to assess fatigue as the dependent factor in our association model [29,30]. The FSS is a self-reporting questionnaire that assesses fatigue severity in daily life and consists of nine statements about patients’ perceived fatigue, in which each item is rated on a 7-point Likert scale [31]. The recall period is not specified for the FSS [31]. The FSS has shown excellent internal consistency for patients with stroke (Cronbach’s alpha > 0.90) and excellent test–retest reliability for stroke patients (intraclass correlation coefficient = 0.93) [29]. An FSS score ≥ 4 is interpreted as indicative of fatigue [31,32].

### 2.5. Measuring Cardiorespiratory Fitness

As the central determinant of interest to quantify CRF, we measured the maximal exercise capacity by performing a ramp exercise test on a Jaeger cycle ergometer under continuous blood pressure measurement, ECG, and breath-by-breath gas analysis [24]. CRF refers to the ability to transport and use oxygen and is usually expressed as maximal oxygen uptake (V̇O2max) [24].

### 2.6. Measuring Potential Confounding Factors

Confounding was defined as the distortion in the presumed association between PSF and CRF due to factors associated with both PSF and CRF [33]. This study considered the following candidate confounders, which were assumed to be bivariately associated with the dependent variable PSF and the central determinant CRF: age, gender, self-reported level of physical activity, cognitive functioning, depression, anxiety, stroke type, treatment allocation, and comorbidity. Self-reported physical activity was defined as “all bodily movement that is produced by the contraction of skeletal muscle and that substantially increases energy expenditure” [34]. This construct was scored using the Physical Activity Scale for the Elderly (PASE) questionnaire [35]. Cognitive functioning was assessed with the Montreal Cognitive Assessment (MoCA) [36]. The Hospital Anxiety and Depression Scale (HADS) was used to assess self-reported anxiety and depression [37]. Assuming that severity of stroke (i.e., TIA or minor stroke) may influence PSF as well as CRF, stroke type was also considered as a candidate factor. Since the subjects were participating in a randomized controlled trial, treatment allocation was also a possible confounder. Finally, the following comorbidities were included in the analysis as separate possible confounders and were scored as present or absent: history of myocardial infarction, history of peripheral arterial disease, history of psychiatric diseases, epilepsy, chronic obstructive pulmonary disease (COPD) or asthma, and type 1 or 2 diabetes mellitus.

### 2.7. Procedure

All variables were assessed at one year post-stroke. One observer, blinded to the treatment allocation and not involved in the data analysis, performed the measurements and administered the questionnaires. A pulmonary function technician, blinded to the treatment allocation, conducted the V̇O2max test.

### 2.8. Statistical Analysis

First, bivariate regression analyses were conducted with the FSS score as the dependent variable and V̇O2max as the independent variable to test our primary hypothesis. A standardized beta coefficient was calculated. Subsequently, the effect of CRF on PSF was investigated while controlling separately for the abovementioned candidate factors of age, gender, physical activity level, cognitive functioning, depression, anxiety, stroke type, treatment allocation, and co-morbidity as potential confounders. If the regression coefficient of V̇O2max changed beyond 15% after adding a variable to the model, the added covariate was considered to be a significant confounder [33,38]. Finally, a multivariate regression model was applied, based on all significant confounding factors, to determine the unique explained variance between PSF and CRF. A 2-tailed significance level of 0.05 was used for all analyses.

## 3. Results

### 3.1. Patient Characteristics

Table 1 presents the characteristics of the 119 included patients with a TIA or minor stroke, 59% of whom were men. The mean age was 65 years (SD = 10). The groups of TIA and minor stroke patients were almost the same size. The mean score on the FSS was 3.9 (SD 1.5). Fifty-four (48%) patients scored 4 or higher on the FSS, which may be indicative of clinically significant fatigue. The mean V̇O2max was 22.4 mL/kg/min, and only seven of the 64 (11%) male patients and two of the 39 (5%) female patients had a V̇O2max above the cut-off point for poor or very poor V̇O2max based on their gender and age [24].

### 3.2. Bivariate Associations

Table 2 presents the bivariate relations between the FSS score and each of the independent variables. V̇O2max was significantly negatively associated with FSS (ß = −0.061, SE 0.022, *p* = 0.007). Furthermore, PASE (−0.004, SE 0.002, *p* = 0.023), HADS depression (ß = 0.194, SE 0.035, *p* < 0.000), HADS anxiety (ß = 0.224, SE = 0.037, *p* < 0.000), treatment allocation (ß = −0.943, SE = 0.269, *p* < 0.001), and the presence of comorbidity (i.e., COPD/asthma; ß = −1.294, SE = 0.617, *p* = 0.038) were significantly associated with FSS. Age, gender, cognitive functioning, stroke type, and the other types of co-morbidities were not statistically significantly associated with FSS.

### 3.3. Confounding Factors and Multivariate Association

Table 3 presents the adjusted regression coefficient of the association between FSS and V̇O2max after correcting for the significant candidate confounders of physical activity level, depression, anxiety, COPD/asthma at one year, and treatment allocation. The addition of HADS depression to the model resulted in a proportional decrease of 50% of the regression coefficient of FSS (ß = −0.030, SE = 0.022, *p* = 0.177). The addition of HADS anxiety proportionally reduced the regression coefficient of FSS by 28% (ß = −0.044, SE = 0.020, *p* = 0.028). After controlling for HADS depression and HADS anxiety separately, the association between V̇O2max and FSS was no longer significant, whereas no significant changes resulted from controlling for the other covariates of PASE, treatment allocation and COPD/asthma. In a multivariate regression model, which included both HADS depression and HADS anxiety, the association between V̇O2max and FSS became non-significant.

## 4. Discussion

In the present study, we found that the self-reported impact of PSF measured with FSS at one year post-stroke has a weak to moderate but significant association with CRF measured as V̇O2max, explaining about 30% of the variance in subjects with TIA and minor stroke. However, this association was significantly confounded by the factors of depression and anxiety. The above findings suggest that the association between PSF and CRF is weak at best, and that the relationship between PSF and CRF cannot be interpreted without considering depression and/or anxiety as potential confounding factors. This finding also suggests that in subjects with PSF showing low performance on a V̇O2max test and underlying feelings of anxiety and depression should be considered. Importantly, nearly half of the patients with TIA or minor stroke in the current study met the criteria for clinically significant fatigue one year post-onset, as measured with the FSS, which is a percentage similar to that of patients with major stroke [39,40]. In addition, 89% of the male and 95% of the female patients had V̇O2max values below the expected age- and gender-matched values found in healthy subjects [24]. This finding further confirms that most stroke subjects even without significant disabilities also suffer from PSF and a significantly reduced CRF when compared to healthy subjects.

To the best of our knowledge, this is the first study to investigate the association between PSF and CRF in patients with TIA and minor stroke. A recent systematic review with meta-analysis from 2023, which included 32 studies and three study protocols (N = 4721), found a weak but significant association between higher PSF and impaired fitness [7]. The authors suggested that CRF can be a protective factor against PSF. However, this review pooled studies with subjects with major stroke. In addition, the included trials were small-sampled, did not specifically target fatigue as a primary measurement of outcome, and used different measurements of fatigue measured at different time points post-stroke. In addition, the authors suggested that the weakness of the association may have been due to additional factors that were not included, such as depression [7]. In contrast, our study focused on subjects with TIA or minor stroke. These subjects are discharged rapidly from the hospital and, if treated, they receive secondary prevention such as control of blood pressure and diabetes and life-style education. With that, subtle impairments as PSF, anxiety and depression may not be identified if these symptoms are not specifically screened in these subjects.

The results from the current study have consequences for future research among and treatment of patients suffering from PSF. When treating PSF in patients with TIA or minor stroke, physicians should consider not only measuring the CRF but also assessing the possible presence of anxiety and depression. In addition, little is known about how to supervise patients with an anxiety disorder and/or depression during a cardiopulmonary exercise test, and more research is needed to determine how to perform a successful cardiopulmonary exercise test with patients with an anxiety disorder and/or depression [41,42]. Furthermore, as it is not clear which treatment modality is most effective, future randomized controlled trials are needed to investigate the effectiveness of multidisciplinary interventions targeting the improvement of PSF and CRF, as well to reduce depression and anxiety in patients with TIA and stroke. Such interventions must be administered by a multidisciplinary team paying attention to both physical and psychological functioning.

A major difficulty in PSF research is the lack of consensus on how to assess and diagnose PSF [43]. A large variety of patient-reported outcome measures (PROMs) to measure PSF are being used, none of which have been developed specifically for patients with TIA or stroke [43]. In the current study, we measured PSF with a unidimensional questionnaire, the FSS, which measures the impact on an individual’s daily functioning [31]. Although unidimensional measures have the advantage of being brief and easily administered, they are limited in their ability to capture the full range of fatigue-related symptoms experienced by individuals [44]. In contrast, multidimensional fatigue questionnaires assess a broad range of domains in which fatigue may manifest itself, including physical, affective, cognitive, and functional symptoms [45]. Recently, the Fatigue Impact Scale (FIS) was validated as a valid and reliable multidimensional questionnaire for patients with stroke, measuring the effect of fatigue on three domains of daily life: cognitive functioning, physical functioning, and psychosocial functioning [46]. A disadvantage of multidimensional questionnaires is that they usually contain numerous questions, which makes completing the questionnaire a greater burden for patients. Although validation for TIA and stroke patients is still required, a potential alternative that is short and complete at the same time could be the use of the fatigue questionnaires of the PROMIS system. PROMIS is a patient-reported system based on item-response theory that uses a continuous scale to efficiently measure patients’ health status for different domains of health [47]. The use of computer-adaptive testing (CAT) allows the system to query patients based on their previous responses until a pre-specified level of precision is reached, typically within three to four questions, about aspects of fatigue. This method significantly reduces the burden on the patients who complete the questionnaire [48,49]. Since there has been only one validation study showing that PROMIS fatigue measures are clinically valid for six common chronic conditions [50], more research is needed to further psychometrically validate the PROMIS CAT as a more efficient way to measure PSF. Furthermore, we recommend choosing a questionnaire that reflects the specific aspect of fatigue in the research question. For example, the FSS measures the severity of fatigue, while the effect of fatigue on domains of daily living is measured with the FIS.

There are some limitations to this study. First, we may have missed specific fatigue symptoms because PSF was measured with a unidimensional questionnaire. Second, our result is not generalizable to the whole group of stroke patients. We recommend that future studies include both patients with TIA and those with stroke, with and without physical disabilities. Third, the present study included patients from a randomized clinical trial in which they received a one-year exercise intervention or usual care. Patients in the intervention group received more guidance toward achieving an active lifestyle, and more exercise training to achieve a better CRF, than patients in the control group, which may have affected the association between PSF and CRF. However, this neutral trial found no significant difference between the intervention and control groups on the primary outcome of global cognition, as assessed by the Montreal Cognitive Assessment (MoCA), or on CRF [19]. Because this trial did found a significant between-group difference in fatigue (in favor of the experimental group at 12 months), we included treatment allocation in the analysis. Treatment allocation turned out not to be a confounder in the association between PSF and CRF. The present study was part of a neutral trial [19]. Patients allocated to the experimental intervention received CRF, whereas the control group received motivational interview-based counselling that focused on the attainment of secondary prevention and the improvement of both lifestyle factors and physical activity. In light of the absence of any between-group differences, treatment allocation was not a significant confounder in our current association model. Fourth, we may have missed possible confounders because they were not included in the MoveIT trial. For example, despite the possible association between sleep disturbances and PSF [28], this determinant was not measured and, therefore, could not be included in our analyses.

## 5. Conclusions

This study found that the association between PSF and CRF must be interpreted with caution, as depression and/or anxiety are confounding factors in patients with TIA and minor stroke. This study shows the importance of measuring depression and anxiety together with PSF and CRF in patients with TIA and minor stroke to ensure that subtle impairments are detected and can be treated. This result also provides an indication that future randomized controlled trials and treatments should focus not only on improving CRF and PSF, but also target depression and anxiety in patients with TIA and minor and major stroke.

## Figures and Tables

**Table 1 brainsci-13-00561-t001:** Patient characteristics at one year post-onset (N = 119).

Variable	Mean (SD)
Gender: male/female ^a^	70/49
Age (y) (range 44–86)	64.8 (9.7)
NIHSS (median) interquartile range	0 (1)
FSS (range 1–7)	3.8 (1.5)
V̇O2max (mL/kg/min)	22.4 (6.5)
PASE (range 0–361)	148.3 (82.0)
MoCA (range 0–30)	25.9 (2.9)
HADS depression (range 0–21)	4.3 (3.6)
HADS anxiety (range 0–21)	4.5 (3.3)
Stroke type (TIA/minor stroke) ^a^	60/59
Time post stroke in days	427 (49)
Comorbidity	
History of myocardial infarction ^a^	5
History of peripheral vascular disease ^a^	3
History of psychiatric disease ^a^	9
Epilepsy ^a^	4
COPD/asthma ^a^	6
Type 2 diabetes mellitus^a^	14

^a^ Expressed as number of patients. Abbreviations: NIHSS, National Institutes of Health Stroke Scale; FSS, Fatigue Severity Scale; V̇O2max, maximal oxygen consumption; PASE, Physical Activity Scale for the Elderly; MoCA, Montreal Cognitive Assessment; HADS, Hospital Anxiety and Depression Scale; TIA, transient ischemic attack; COPD, chronic obstructive pulmonary disease.

**Table 2 brainsci-13-00561-t002:** Bivariate association between FSS and candidate determinants at one year (N = 119).

Determinant	ß Value of the Determinant	Standardized ß Value of Determinant
V̇O2max	−0.061 (0.022) *	−0.266
Gender	0.303 (0.290)	0.099
Age	0.003 (0.015)	0.020
PASE	−0.004 (0.002) *	0.226
MOCA	0.003 (0.050)	0.005
HADS depression	0.194 (0.035) *	0.470
HADS anxiety	0.224 (0.037) *	0.497
Stroke type (TIA/minor stroke)	0.183 (0.283)	0.061
Treatment allocation	−0.943 (0.269) *	−0.317
Comorbidity		
History of peripheral vascular disease	0.835 (0.874)	0.091
History of psychiatric disease	−1.583 (0.529)	−0.274
Epilepsy	−0.117 (0.877)	−0.133
COPD/asthma	−1.294 (0.617) *	−0.196
Type 2 diabetes mellitus	−0.063 (0.428)	−0.014

* *p* < 0.05; the numbers in parentheses represent standard error (SE). Abbreviations: V̇O2max, maximal oxygen consumption; PASE, Physical Activity Scale for the Elderly; MoCA, Montreal Cognitive Assessment; HADS, Hospital Anxiety and Depression Scale; TIA, transient ischemic attack; COPD, chronic obstructive pulmonary disease.

**Table 3 brainsci-13-00561-t003:** Bivariate multilevel and multivariate regression model to test the effect of confounders on the predictive value of V̇O2max for fatigue at one year (N = 119).

Variable in the Model	ß Value of CandidateConfounder	ß Value of V̇O2max	Standardized ß Value of V̇O2max	Proportional Change in V̇O2max
Bivariate regression model				
V̇O2max		−0.061(0.022)		
HADS depression	0.164 (0.040)	−0.030 (0.022)	−0.129	−50%
HADS anxiety	0.204 (0.038)	−0.044 (0.020)	−0.193	−28%
Multivariate regression model				
HADS depression + HADS anxiety		−0.033 (0.021)	−0.143	

The numbers in parentheses represent standard error (SE). Abbreviations: V̇O2max, maximal oxygen consumption; HADS, Hospital Anxiety and Depression Scale.

## Data Availability

Data available upon request.

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
