# Peer review of "Is Cardiorespiratory Fitness Independently Associated with Fatigue in Patients with Transient Ischemic Attack or Minor Stroke?"

_brainsci, 2023, doi:10.3390/brainsci13040561_

Round 1
Reviewer 1 Report
Thank you for submitting this paper. I have made a number of comments and edits on the paper itself (attached). Although I agree with your conclusions that fatigue in these patient cohorts must be carefully assessed to eliminate possible confounders, I am not sure that in its present form this paper adds to the current literature. As I have noted many of your references are > 20 years old and as I have stated unless seminal should be updated.

Reviewer 2 Report
Thank you for the opportunity to review this manuscript
- Please include the age range, since cardiorespiratory fitness and fatigue may vary according to age; the study included patients who are at least 18 years, and the mean age was 64.8 (9.7). I suggest making an analysis related to age.
- A major concern is about the included subjects of this study, the homogeneity of patients’ regarding the received intervention is questionable since the control group received no intervention, an important factor that may influence the outcomes of the studied association, and consequently may negatively contribute to the internal validity of the study.
- The study concluded that “These results suggest that the association between PSF and CRF is weak and significantly confounded by the factors depression and anxiety in patients with TIA or minor stroke”.
I think that drawing a conclusion in the presence of the mentioned confounding variables remains difficult. Please clarify what would be the implications of these results.
Reviewer 3 Report
Deijle et al are presenting an excelent study adressing evaluating the association between fatigue and cardiorespiratory fitness in patients with TIA an minor stroke?. The manuscript is clear and easy to follow. I do have some minor points:
1. Replace TIA by transient ischemic attack in the title
2. Present the flowchart of patient inclusion (from the origina cohort to the population included in the study, including lost to follow-up patients)
3. It is a surprise that treatment allocation did not modulate significantly to the association between PSF and CRF. I am not familiarized with the intervention per se (duration, dosage, adherence ...), but this should be discussed.
Round 2
Reviewer 1 Report
I thank the authors for the changes that they have made, which I believe greatly improves the readability of the paper.
I have indicated some very minor edits which I believe with also further improve this paper.

Reviewer 2 Report
Thank you for taking my comments into consideration.